# Tetraethylammonium Acetate and Tetraethylammonium Bromide-Based Deep Eutectic Solvents as Thermodynamic CO₂ Gas Hydrate Inhibitors

**Vinayagam Sivabalan** [1] , **Nurasyikin Hasnor** [1,2,3] , **Bhajan Lal** [1,2,*] , **Zamzila Kassim** [4] **and Abdulhalim Shah Maulud** [1,5]

[1] Chemical Engineering Department, Universiti Teknologi PETRONAS, Perak 32610, Malaysia; vinayagam_19000196@utp.edu.my (V.S.); nhasnor@mcdermott.com (N.H.); halims@utp.edu.my (A.S.M.)
[2] CO2 Research Centre (CO2RES), Universiti Teknologi PETRONAS, Perak 32610, Malaysia
[3] McDermott Asia Pacific Incorporation, Kuala Lumpur 50250, Malaysia
[4] Group Research & Technology, PETRONAS Research Sdn Bhd, Lot 3288&3289, Off Jalan Ayer Itam, Kajang 43000, Selangor, Malaysia; zamzila_kassim@petronas.com
[5] Center of Contaminant Control & Utilization (CenCoU), Universiti Teknologi PETRONAS, Perak 32610, Malaysia
* Correspondence: bhajan.lal@utp.edu.my

**Abstract:** The thermodynamic gas hydrate suppression behavior of four Deep Eutectic Solvents (DESs) was evaluated in this paper. The mixtures of Hydrogen Bond Acceptors (HBA), Tetraethylammonium Acetate (TEAAC), and Tetraethylammonium Bromide (TEAB) with Hydrogen Bond Donors (HBD), Mono-Ethylene Glycol (MEG), and Glycerol were used to make the DES. The DESs were made at a 1:7 molar ratio for the combinations of TEAAC:MEG, TEAAC:Glycerol, TEAB:MEG, and TEAB:Glycerol. The Hydrate Liquid-Vapor Equilibrium (HLVE) data for CO₂ were evaluated through the T-cycle method at different temperature (273.15–283.15 K) and pressure (2–4 MPa) conditions in the presence and absence of 5 wt % aqueous DES solutions. The inhibition effects showed by the DESs, including average suppression temperature ($\Delta\bar{T}$) and gas hydrate dissociation enthalpies ($\Delta H_{diss}$), were also calculated. The average suppression temperature values of the DESs ranged between 0.4 and 2.4, with the highest inhibition to lowest inhibition order being TEAB:Glycerol > TEAB:MEG > TEAAC:Glycerol > TEAAC:MEG. A comparison of the DES with conventional Thermodynamic Hydrate Inhibitors (THIs) showed that studied Deep Eutectic Solvents had better gas hydrate inhibition. The results proved that DES has the potential to be one of the promising alternatives in gas hydrate inhibition.

**Keywords:** gas hydrates; deep eutectic solvents; HLVE data

## 1. Introduction

Gas hydrates are stable inclusion compounds made up of a network of water molecule cages that act as a host that trap light hydrocarbon as guest statistical analysis on experimental Hydrate Liquid-Vapor Equilibrium (HLVE) data under high pressure and low-temperature conditions during deep-water energy exploration and recovery [1–7]. The risk of gas hydrate formation is highly prevalent and poses a significant operational and safety challenge [8,9]. When the pipeline is clogged with gas hydrates, the explosion of pipes can occur, which will pose a threat to safety and environment and lead to economic loss [10–13].

Although many methods are handled to prevent hydrate formation, the chemical inhibition method is the most prevalent [14]. The conventional way of preventing gas hydrate formation is to

inject Thermodynamic Hydrate Inhibitors (THIs) into pipelines, thereby shifting hydrate formation conditions to higher pressure and lower temperature regions [15–19]. However, there are a few drawbacks associated with the conventional gas hydrates inhibitor, especially on the volatility of the chemical. Thus, researchers have been working towards Ionic Liquids (ILs) as gas hydrate inhibitors but currently looking for an alternative as the ILs itself have their disadvantages [1,4,6,15,17,18,20–22]. ILs need a high purity, are expensive, require a large number of salts and solvents for complete exchange of anions, and have poor biodegradability. These drawbacks can be overcome with the use of Deep Eutectic Solvent (DES), which is known for its cheap, non-toxic, and biodegradable nature.

DES is an analog of ILs, which possesses a few similar characteristics of an ionic liquid. DES is usually in liquid form and is usually formed by mixing a quaternary ammonium salt with hydrogen bond donors [23]. Generally, a eutectic system is a mixture of components having the lowest melting point through virtue of specific proportions. Without covalent or ionic bonds, these components interact merely through intermolecular forces [24]. Detailed works comparing DES and ILs have been done, and DESs have been slowly replacing ILs in various fields [23–33]. DESs have great potential to be effective and eco-friendly Thermodynamic Hydrate Inhibitors (THIs) because DESs have functional groups capable of forming hydrogen bonds with water. DESs also have high biodegradability and low toxicity, besides exhibiting negligible vapor pressure [34–36]. DES is known for its cheap, non-toxic, and biodegradable characteristic, which is also characterized by a significant depression of low melting point. Four types of DESs can be formed, namely, (1) quaternary ammonium salt + metal chloride, (2) quaternary ammonium salt + metal chloride hydrate, (3) quaternary ammonium salt + hydrogen bond donor, and (4) metal chloride hydrate + hydrogen bond donor [25,31].

Nature showed by Choline Chloride, Propanedioic Acid, 3-Phenyl Propionic Acid, Itaconic Acid, and 3-Mercaptopropionic Acid-based DESs, as potential shale inhibitors in water-based drilling fluids, is further evidence of the DESs to be efficient THIs [37]. In the research work by Aissaoui et al. [38], DESs based on Choline Chloride displayed significant absorption of $H_2O$ from natural gas down to the concentrations specified for pipelines to avoid hydrate formation. As DESs are generally formed by mixing Hydrogen Bond Acceptors (HBA) and Hydrogen Bond Donors (HBD), there is a significant margin to be exploited in making the greener gas hydrate inhibitors. Many substances have been tested to be potential gas hydrate inhibitors but have displayed either biodegradability or performance issues. However, mixing potential compounds to make DESs may be a beneficial act. Quaternary Ammonium Salts like Tetramethylammonium Bromide (TMAB), Tetraethylammonium Bromide (TEAB), Tetraethylammonium Acetate (TEAAC), and Tetraethylammonium Iodide (TEAI) have shown good $CO_2$ hydrate inhibiting behavior in recent research works.

Along with that, several HBDs, such as Ethylene Glycol, Diethylene Glycol, Glycerol, and Urea, have shown hydrate inhibition behavior [14,34]. Deep Eutectic Solvents have tunable properties that can be exploited for specific usage as required [39]. The idea behind this work is to synthesize Deep Eutectic Solvents from the pure compounds that behave as thermodynamic gas hydrate inhibitors.

The preparation of DES is simple by the mixing of Hydrogen Bond Donor (HBD) and Hydrogen Bond Acceptor (HBA) at a specific temperature in two ways: (1) when the HBD and HBA are mixed, the lower melting point constituent begins to melt, and then the remaining compound that has a high melting point is put into the liquid, and the mixtures are melted collectively, and (2) when both constituents are mixed and melt together, since the first work of Abbot et al. 2003 [40]. It is supposed to be noticed that the homogeneous mixture of liquid is the ending stage of the procedure of Deep Eutectic Solvent preparation. Besides, an easy way to confirm the formation of DES besides the homogeneity of the mixture is to freeze-thaw the homogenous mix of compounds and observe any heterogeneity. If the combination remains homogeneous, then it can be concluded that the deep Eutectic Solvent has been formed.

In this work, DESs from TEAAC:Mono-Ethylene Glycol (MEG), TEAAC:Glycerol, TEAB:MEG, and TEAB:Glycerol were formed at 1:7 molar ratios and tested for their $CO_2$ hydrate inhibition performance at various operating conditions with and without 5 wt % aqueous DES solutions.

Selected experimental conditions were in the range of 273 K to 283 K for temperature and 2 MPa to 4 MPa for pressure. The Hydrate Liquid-Vapor Equilibrium (HLVE) curves and average suppression temperature were obtained to understand the extent of thermodynamic suppression offered by the selected compounds. The dissociation enthalpies ($\Delta H_{diss}$) for $CO_2$ gas hydrate were also considered in this work.

## 2. Methodology

### 2.1. Materials

All the Quaternary Ammonium Salts (QAS) and Hydrogen Bond Donors (HBD) used for this study are purchased from Sigma-Aldrich (M) Sdn. Bhd. (Petaling Jaya, Selangor, Malaysia). Linde Malaysia Sdn Bhd (Petaling Jaya, Selangor, Malaysia) delivered the $CO_2$ gas used for the hydrate study. Bhd. Deionized water was used for the dilution of the chemicals up to the required concentrations.

In this project work, the agitation strategy was used to formulate DES. Firstly, the Quaternary Ammonium Salt and Hydrogen Bond Donor were mixed at a 1:7 molar ratio. Then, the mixture was stirred for one hour at 650 rpm in a room condition. If the blend was miscible, a homogenous transparent liquid could be formed. The formation of DES could be seen if a transparent uniform mixture was obtained after settling down the mixture to ambient temperature for 15 min. The chemicals used are tabulated in Table 1.

**Table 1.** List of chemicals.

| Chemical Name | Chemical Formula | Chemical Structure |
|---|---|---|
| Tetraethylammonium Acetate Tetra-Hydrate | $(C_2H_5)_4N(OCOCH_3) \cdot 4H_2O$ | |
| Tetraethylammonium Bromide | $(C_2H_5)_4N(Br)$ | |
| Glycerol | $C_3H_8O_3$ | |
| Mono-Ethylene Glycol | $(CH_2OH)_2$ | |
| Carbon Dioxide | $CO_2$ | |

### 2.2. Experimental Procedure

#### 2.2.1. Characterization Using Fourier Transform Infrared Spectroscopy (FTIR)

Once the homogenous transparent liquid could be formed, FTIR equipment available in Centralised Analytical Lab (CAL), Universiti Teknologi Petronas (Seri Iskandar, Perak, Malaysia) was used to identify the functional group of the formulated DESs. The interaction between the Quaternary Ammonium Salt (QAS) and Hydrogen Bond Donor (HBD) when DES was formed could be analyzed based on the changes observed in the peaks. The intensity of the peaks changed when there was a change in chemical bonding. All analysis done in this part was based on the interpretation of infrared spectra by John Coates [41]. According to the literature, there are a few steps to analyze the infrared spectrum, as tabulated in Table 2.

**Table 2.** Steps to analyze FTIR.

| The Wavelength Range for Different Purposes of Testing | | |
|---|---|---|
| No | Purpose | Range ($cm^{-1}$) |
| 1 | Testing for organics and hydrocarbons | 3200–2700 |
| 2 | Testing for hydroxyl or amino groups | 3650–3250 |
| 3 | Testing for carbonyl compounds | 1850–1650 |
| 4 | Testing for unsaturation | 1670–1620 |
| 5 | Testing for aromatics | 1615–1495 |
| 6 | Testing for multiple bonds | 2300–1990 |

The FTIR analysis was performed with Thermo Scientific™ Nicolet™ iS5 FTIR Spectrometer that has Thermo Scientific iD7 ATR accessory with the diamond crystal. The Potassium Bromide (KBr) pellet method was implemented. Background scanning was performed to recognize the presence of air impurities, such as carbon dioxide peak for every sample. The spectrometer with a 2 $cm^{-1}$ resolution was used to generate spectra with patterns that provide structural insights in the wavenumber range of 4000–400 $cm^{-1}$ with 32 number of scans. The analysis of the DES samples available in liquid state was done at ambient temperature. Results of TEAAC:MEG, TEAAC:Glycerol, TEAB:MEG, and TEAB:Glycerol were compared to the FTIR of pure components TEAB, TEAAC, MEG, and Glycerol available on SpectraBase website provided by John Wiley and Sons, Inc. (Hoboken, NJ, USA).

### 2.2.2. Gas Hydrate Inhibition Study

The HLVE data of $CO_2$ hydrate was determined through the T-cycle method, which is a study approach using a constant volume thermodynamic cycle. The reactor cells were thoroughly cleaned before the experiment to avoid the effect of any impurities. The 15 mL of sample DES solutions were injected into each cell. The 6-cell reactor was then chilled to the desired experimental temperature. Once the desired temperature was attained, the cells were purged three times and vacuumed to dispose of any miscellaneous gas traces inside them. $CO_2$ gas was then injected into the cells up to the required pressure range. The range of pressure for experiments of $CO_2$ + DES gas hydrates was set between the range of 2 and 4 MPa as typically encountered in transmission pipelines. The 6-cell reactor equipped with a 1.8 cm diameter glass ball in each cell was put into a rocking mechanism that provided adequate mixing between gas and liquid phase. The stirring was essential to break the liquid water interface boundary during the hydrate formation. The sudden significant pressure drop observed in the recorded data was the primary indicator for the formation of gas hydrates. When there was no further pressure drop in the system, hydrates were deduced to be formed fully, and the system was heated at a constant rate of 0.25 °C/min [15]. Figure 1 shows the schematic diagram of the experimental apparatus, which consists of six separate units of identical rocking cells controlled by a built-in programmable logic controller programmed by Dixson FA Engineering Sdn. Bhd (Shah Alam, Selangor, Malaysia). The details of the equipment can be found in previous works in the literature [42,43].

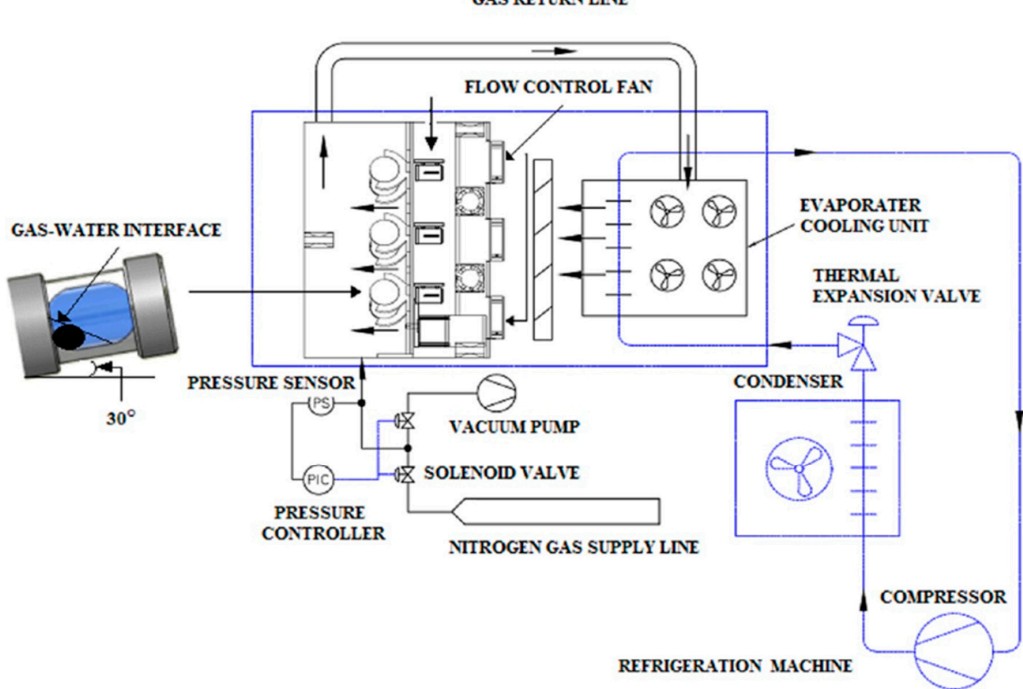

**Figure 1.** Experimental schematic diagram of rocking cell hydrate reactor [42].

### 2.3. Average Suppression Temperature (T̄) and Dissociation Enthalpies ($\Delta H_{diss}$)

The average suppression temperatures (T̄) of the DESs were calculated to determine the thermodynamic $CO_2$ hydrates suppression capability of the Deep Eutectic Solvents tested in this work. The following equation (1) was implemented to determine the suppression temperature ($\Delta T$), as used by Qasim et al. [15].

$$T = \frac{\Delta T}{n} = \frac{\sum_{i=1}^{n}\left(T_{0,pi} - T_{1,pi}\right)}{n} \tag{1}$$

where $T_{0,pi}$ represents the equilibrium temperature of studied gas in a blank sample, while $T_{1,pi}$ is the equilibrium temperature of $CO_2$ gas in the presence of DES. The values of both dissociation temperatures should be attained at the same pi. The $n$ denotes the number of pressure points taken into consideration for the study, which was in the range of 2 to 4 MPa.

The gas hydrate dissociation enthalpies denoted as $\Delta H_{diss}$ could be determined by using the commonly used Clausius–Clapeyron equation, according to Qasim et al. [15]. The HLVE data obtained from experiments were plotted as $\ln P$ against $1/T$, and the slope was considered as the left-hand side of the Equation (2).

$$\frac{d \ln P}{d\frac{1}{T}} = \frac{\Delta H_{diss}}{zR} \tag{2}$$

where $P$ and $T$ denote the equilibrium pressure and temperature, respectively, $R$ represents the universal gas constant. The z is the compressibility factor of the gas involved in the study, while $\Delta H_{diss}$ signifies the enthalpy of gas hydrates dissociation. The average suppression temperature (T̄) and dissociation enthalpies ($\Delta H_{diss}$) of the formed DESs were analyzed for further understanding of the gas inhibition mechanism.

## 3. Results

### 3.1. Materials Characterization

Several combinations of QAS and HBD, including TEAAC, TEAB, Glycine, and Tetraethylammonium Iodide with Glycerol and MEG, were tested for this research work. However, this paper only discussed

DES formed using a 1:7 molar ratio of TEAAC:Glycerol, TEAAC:MEG, TEAB:Glycerol, and TEAB:MEG. The formation of a homogenous mixture under agitation strategy without the dilution of water was considered the final stage of DES formation in this work. Figure 2 below shows the DES formed in respective beakers.

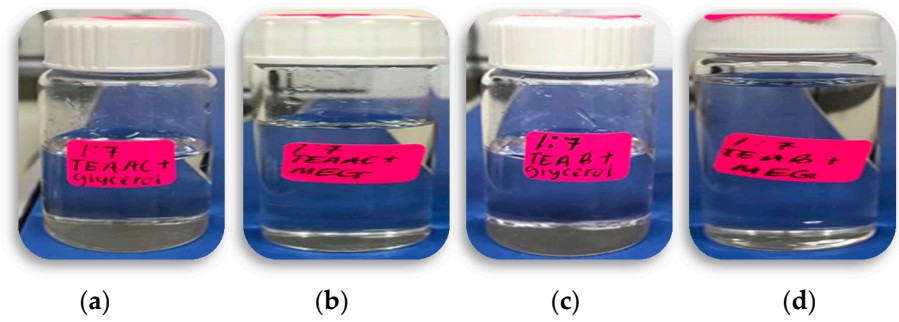

(**a**)      (**b**)      (**c**)      (**d**)

**Figure 2.** Deep Eutectic Solvents in respective beakers. (**a**) TEAAC:Glycerol, (**b**) TEAAC:MEG, (**c**) TEAB:Glycerol, and (**d**) TEAB:MEG.

The FTIR of each compound was compared with the pure components, as shown in Figures 3–6.

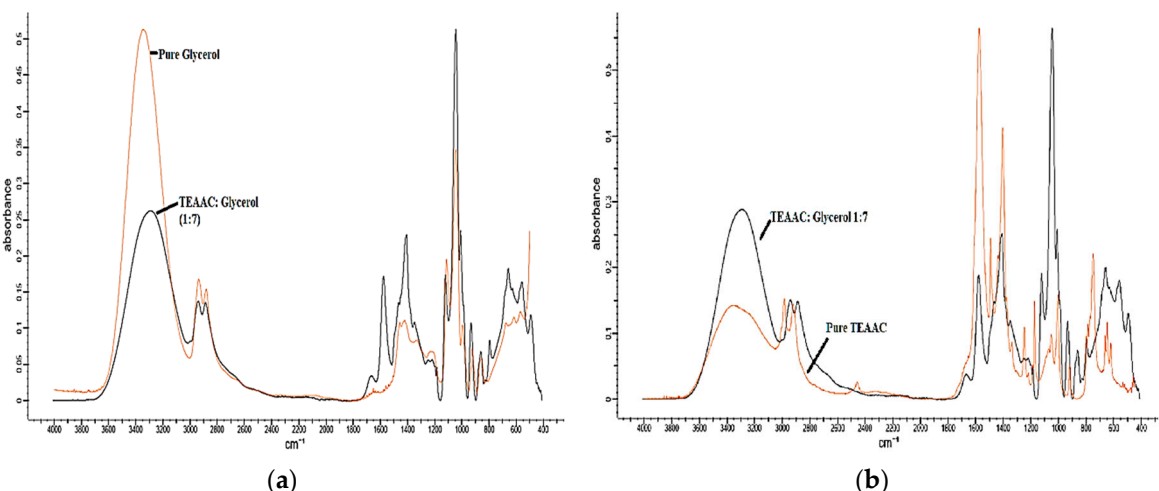

(**a**)      (**b**)

**Figure 3.** The FTIR comparison of TEAAC:Glycerol with (**a**) pure Glycerol and (**b**) pure TEAAC. TEAAC, Tetraethylammonium Acetate.

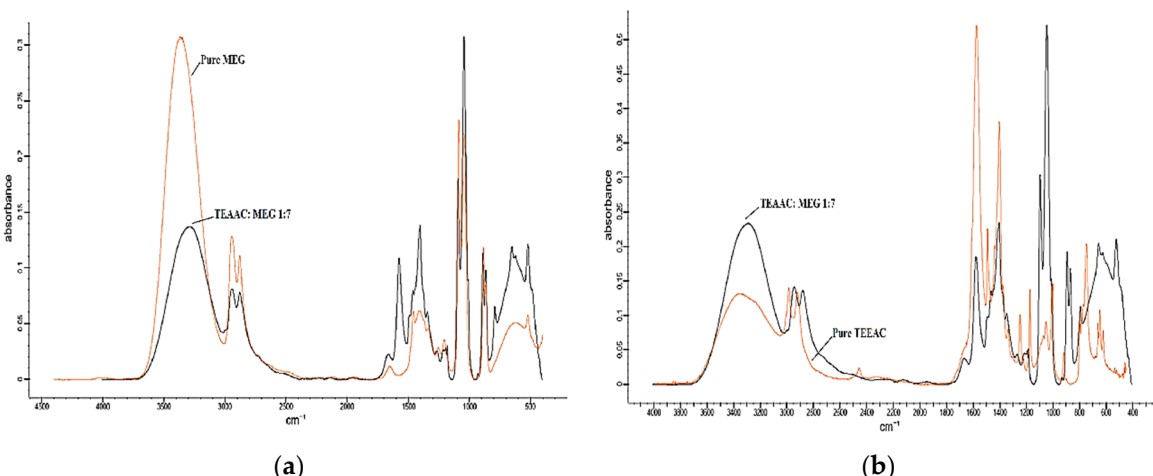

(**a**)      (**b**)

**Figure 4.** The FTIR comparison of TEAAC:MEG with (**a**) pure MEG and (**b**) pure TEAAC. MEG, Mono-Ethylene Glycol.

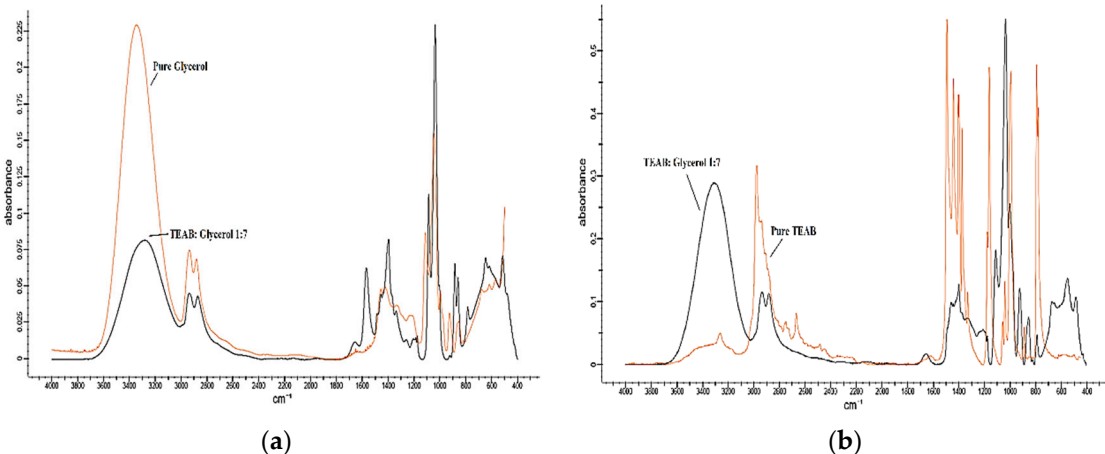

**Figure 5.** The FTIR comparison of TEAB:Glycerol with (**a**) pure Glycerol and (**b**) pure TEAB. TEAB, Tetraethylammonium Bromide.

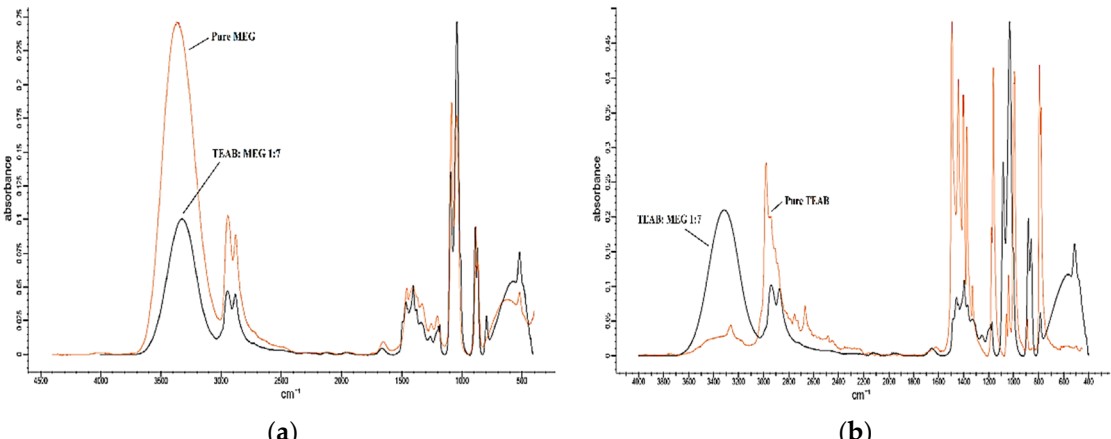

**Figure 6.** The FTIR comparison of TEAB:MEG with (**a**) pure MEG and (**b**) pure TEAB.

Based on Figures 3a and 5a, comparing TEAAC:Glycerol and TEAB:Glycerol with pure Glycerol, it was observed that the FTIR spectrum of DES produced a lower intensity of OH stretch (3400–3200 cm$^{-1}$) in comparison to pure Glycerol spectrum. N–H stretching was also observed to be overlapped with the O–H between 3000 and 3400 cm$^{-1}$, similar to previous works in the literature [44]. The lower intensity was also noticed in the aliphatic stretching vibration between 3000 and 2800 wavelength. By comparing the FTIR spectrum of TEAAC:MEG and TEAB:MEG with the spectrum of pure MEG, as shown in Figures 4a and 6a, respectively, a similar was observed. The intensity of OH stretch and aliphatic stretching vibration was weaker when DES was formed. The reason for this was due to the decrease in the melting point when DES was developed. The reduction in the melting point produced a weaker spectrum of FTIR. However, it was also observed that Glycerol-based DES produced a more vigorous intensity of C–O stretching in comparison to the pure Glycerol spectrum [44]. In other contexts, the MEG-based DES gave almost similar intensity of C–O stretch as the pure MEG.

FromFigures 3b and 4b, it was found that the pure TEAAC spectrum showed a weaker intensity of OH stretch at the wavelength of 3400–3200 when compared to the IR spectrum of TEAAC:Glycerol and TEAAC:MEG. High intensity of C=O stretch at the wavelength of 1650–1590 cm$^{-1}$ shifted to the high intensity of C–O stretch. Besides, Figures 5b and 6b show the pure TEAB to have no OH- stretch in its spectrum, while a peak of OH- stretch could be observed in the DES. The appearance of the OH-stretch indicated that the DES formulated had successfully formed a hydrogen bonding when TEAB was mixed with MEG or Glycerol. Besides, the CH bend from pure TEAB spectrum also shifted to the right where a high intensity of C–O stretch was produced instead. Thus, it could be concluded

from the shifts in the peaks that the change in the chemical bonding occurred when the Deep Eutectic Solvent was formed.

### 3.2. Gas Hydrate Inhibition Study

The gas hydrate inhibiting property of a chemical could be determined from the HLVE data collected during the formation and dissociation of $CO_2$ hydrates in the rocking cell hydrate reactor (Figure 1). The evaluation of thermodynamic gas hydrate suppression considered the findings of equilibrium points where hydrate, liquid, and vapor phases are coexisting simultaneously [15]. The accuracy of the results was validated through measurement and comparison of the data with earlier works of literature [15,45].

In this work, the $CO_2$ equilibrium points were taken by altering the pressure from 2 MPa to 4 MPa in the reactor cells. This range of pressure was determined based on the typical scenario of $CO_2$ dominated oil and gas pipelines in the Energy industry [46]. The equilibrium temperature obtained under this pressure range was between 273.15 and 283.15 K. Table 3 tabulates the HLVE findings of the current work, while Figure 7 attempts to display the results graphically.

**Table 3.** Hydrate Liquid-Vapor Equilibrium (HLVE) data with and without five wt % inhibitors (Deep Eutectic Solvent (DES) of 1:7 molar ratios) for $CO_2$ gas.

| Water | | TEAAC:Glycerol 5 wt % | | TEAAC:MEG 5 wt % | | TEAB:MEG 5 wt % | | TEAB:Glycerol 5 wt % | |
|---|---|---|---|---|---|---|---|---|---|
| T/K | P/MPa | T/K | P/MPa | T/K | P/MPa | T/K | P/MPa | T/K | P/MPa |
| 277.00 | 2.00 | 276.34 | 2.30 | 275.14 | 2.00 | 275.80 | 2.00 | 274.73 | 2.00 |
| 278.80 | 2.50 | 278.30 | 2.70 | 278.50 | 2.50 | 277.45 | 2.49 | 276.40 | 2.50 |
| 280.30 | 3.00 | 279.17 | 3.00 | 280.15 | 3.00 | 277.99 | 3.00 | 277.75 | 3.00 |
| 281.43 | 3.50 | 280.53 | 3.50 | 281.02 | 3.50 | 279.57 | 3.42 | 278.50 | 3.50 |
| 282.52 | 4.00 | 281.42 | 3.80 | 282.39 | 4.00 | 280.36 | 3.84 | 279.90 | 4.00 |

Expanded uncertainties U (T) = ±0.1 K, U (P) = ±0.01 MPa. (0.95 level of confidence). TEAAC and TEAB stand for Tetraethylammonium Acetate and Tetraethylammonium Bromide respectively.

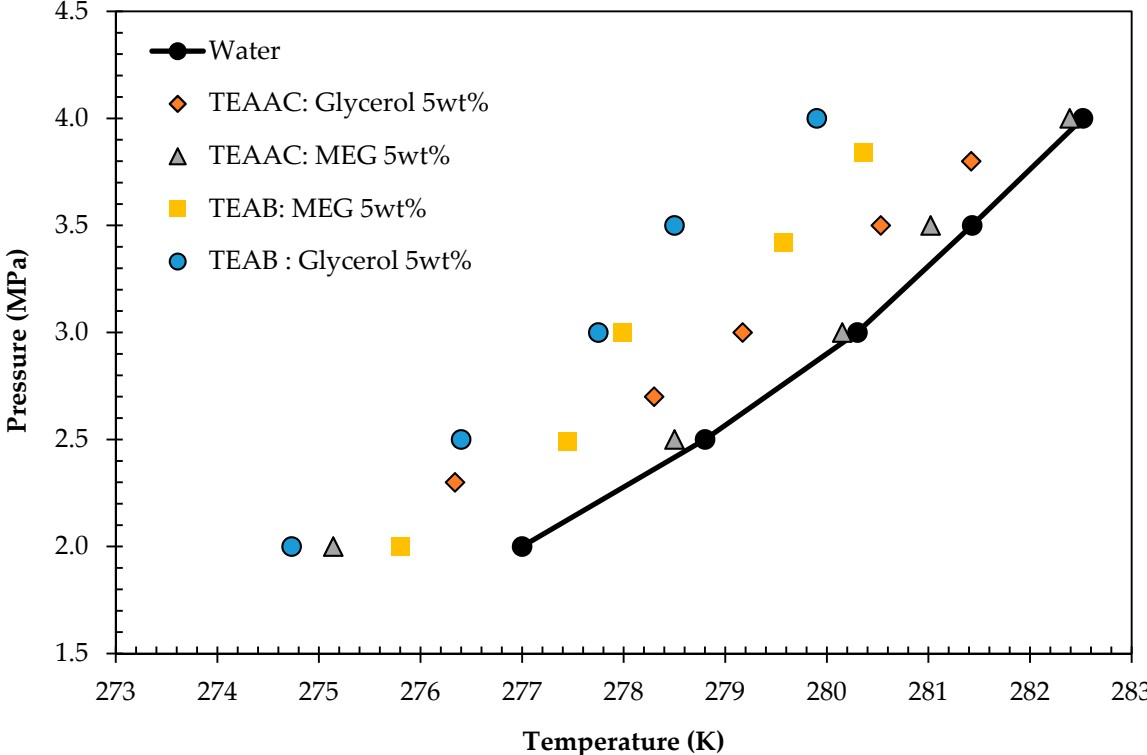

**Figure 7.** Hydrate Liquid-Vapor Equilibrium (HLVE) data of Deep Eutectic Solvent (DES) systems at 5 wt % concentrations.

It was observed from Figure 7 that all four DESs formulated managed to shift the equilibrium boundary of water to the left where lower temperature and higher pressure were required to form hydrates. The shift proved the thermodynamic gas hydrate inhibiting the behavior of DESs. In addition, TEAB:Glycerol showed the maximum change with the lowest equilibrium point at 20 bar and 274.73 K, followed by TEAB:MEG. The gas hydrate inhibition arranged from the highest to lowest would be TEAB:Glycerol > TEAB:MEG > TEAAC:Glycerol > TEAAC:MEG. DESs have great potential to be effective green Thermodynamic Hydrate Inhibitors as they have the functional groups capable of forming hydrogen bonds with water [35,36]. The reliability of the data was validated through statistical analysis, and the results are tabulated in Table 4.

**Table 4.** Statistical analysis of experimental HLVE data.

| Regression Statistics | TEAAC:Glycerol 5 wt % | TEAAC:MEG 5 wt % | TEAB:MEG 5 wt % | TEAB:Glycerol 5 wt % |
|---|---|---|---|---|
| Multiple R | 0.978026 | 0.972272 | 0.972556 | 0.991241 |
| R square | 0.956535 | 0.945313 | 0.945866 | 0.982558 |
| Adjusted R square | 0.942047 | 0.927083 | 0.927822 | 0.976744 |
| Standard error | 0.510676 | 0.483046 | 0.480595 | 0.316228 |
| *Significance F* | 0.003897 | 0.005520 | 0.005435 | 0.000983 |
| Observations | 5 | 5 | 5 | 5 |

Based on the regression analysis, the $R^2$ and adjusted $R^2$ values obtained for the model on the experimental data were generally above 0.90. According to the literature [47,48], the regression statistics can be considered as acceptable with an $R^2$ value > 0.75 and adjusted $R^2$ value > 0.5. In terms of standard error, the values were all below 0.68, signifying the reliability of the data. The significance F values were also below the 0.05 mark, with 0.0055 to be the highest. These values indicated that the set of independent variables were statistically significant.

The superiority of TEAB:Glycerol-based DES over the rest of the DESs could be due to the nature of the pure components [34]. The previous works on TEAB and Glycerol have shown that each of the chemicals has functional hydrate inhibition [11,12,15]. It was proven in this study that the formation of DES using TEAB and Glycerol showed synergistic nature, leading to enhanced results. From the pattern observed, it was agreed that as TEAB was superior to TEAAC and Glycerol was superior to MEG in inhibiting gas hydrate inhibition, the combination of the compounds also showed similar nature to the pure compound. However, it is to be noted that not all DESs will show the same characteristics as some mixtures may show antagonistic behavior.

A fundamental requirement for a chemical to be an effective THI is the potential to form hydrogen bonds with host water molecules, leading to the demand of higher pressure and lower temperature conditions to maintain the hydrate lattice structures. For $CO_2$ hydrates, the THI impact of DES depends on the concentration due to the increased activity of DES in the aqueous phase [15]. However, in this study, the difference caused by the change in concentration was not very significant, albeit shifting the equilibrium to harsher conditions. Table 5 tabulates the HLVE data for 10 wt % of TEAB-based DES. Figure 8 displays the HLVE curve of 5 and 10 wt % of TEAB-based DESs.

**Table 5.** HLVE data with and without ten wt % TEAB-based DES of 1:7 molar ratio for $CO_2$ gas.

| Water | | TEAB:Glycerol 10 wt % | | TEAB:MEG 10 wt % | |
|---|---|---|---|---|---|
| T/K | P/MPa | T/K | P/MPa | T/K | P/MPa |
| 277 | 2.0 | 274 | 1.9 | 274.9 | 2.0 |
| 279 | 2.5 | 276 | 2.5 | 277.5 | 2.8 |
| 280 | 3.0 | 277 | 3.2 | 278.0 | 3.0 |
| 281 | 3.5 | 278 | 3.5 | 279.6 | 3.4 |
| 283 | 4.0 | 279 | 4.0 | 280.0 | 3.9 |

Expanded uncertainties U (T) = ±0.1 K, U (P) = ±0.01 MPa. (0.95 level of confidence).

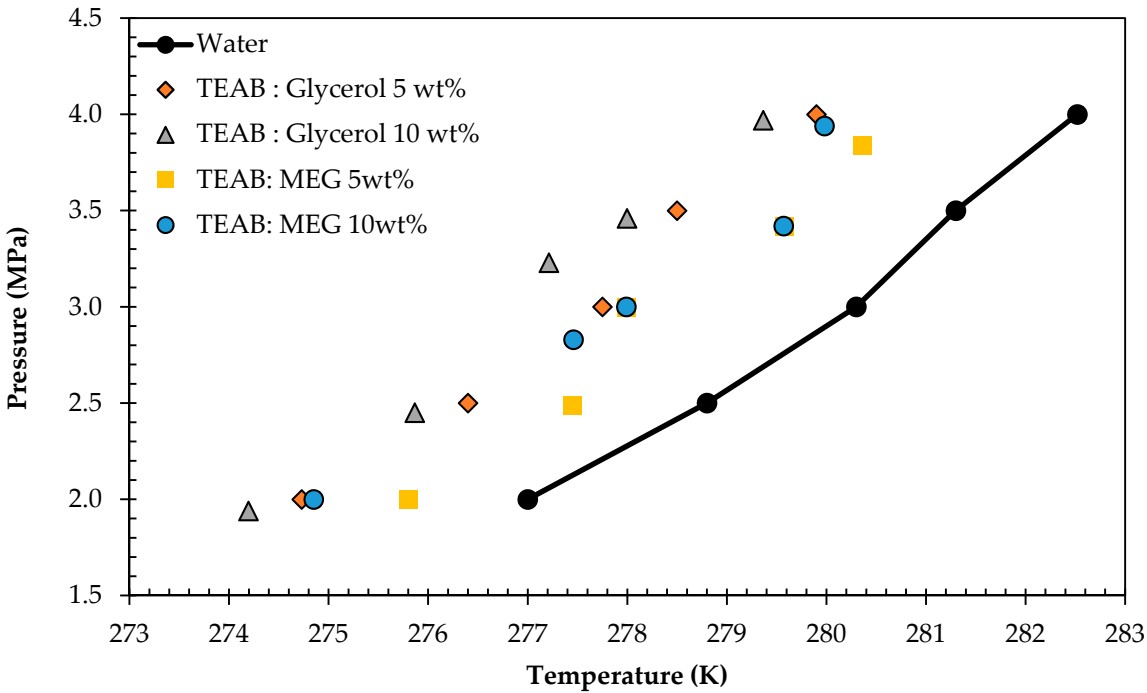

**Figure 8.** HLVE data of TEAB-based DES systems at five wt % and ten wt % concentrations.

As the 10 wt % concentration of TEAB:Glycerol DES was showing good hydrate inhibition, and the authors aimed to suggest a green alternative to currently available THIs, further comparisons were made. Figure 9 displays the HLVE curve of THIs from the current work and literature.

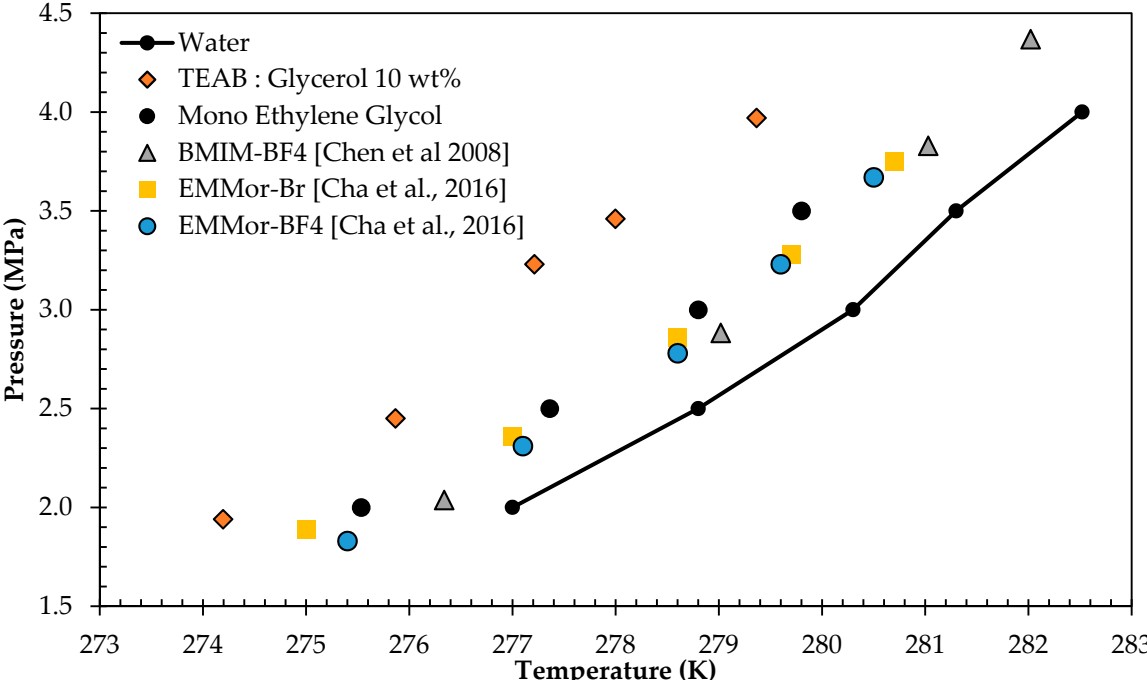

**Figure 9.** Comparison with conventional Thermodynamic Hydrate Inhibitors (THIs).

The phase boundary measurement of 10 wt % TEAB:Glycerol was compared with pure Ethylene Glycol, Ionic Liquid of 1-Butyl-3-Methylimidazolium Tetrafluoroborate (BMIM-BF4), N-Ethyl Methylmorpholinium Bromide (EMMor-Br), and Ethyl-Methylmorpholinium Tetrafluoroborate (EMMor-BF4) [49,50], as shown in Figure 9. The advantages of Ethylene Glycol are that it is recoverable,

reusable, and soluble in water. However, based on the graphical comparison, it could be easily observed that TEAB:Glycerol was a better option for thermodynamic gas hydrate inhibition.

### 3.3. Average Suppression Temperature (T̄)

For a detailed comparison of the impact shown by DES on the $CO_2$ hydrate equilibrium temperature, the authors calculated the average depression temperature ($\Delta T$) demonstrated by the chemicals. Table 6 tabulates the values of average suppression temperature (T̄) of the DESs, while Figure 10 displays the results graphically.

**Table 6.** Average suppression temperature (T̄) of DES systems at different pressure experimental conditions.

|  | TEAAC:Glycerol 5 wt % | TEAAC:MEG5 wt % | TEAB:MEG5 wt % | TEAB:Glycerol 5 wt % |
|---|---|---|---|---|
| P/MPa | $\Delta T/K$ | $\Delta T/K$ | $\Delta T/K$ | $\Delta T/K$ |
| 2.0 | 2.00 | 1.35 | 0.70 | 1.80 |
| 2.5 | 0.83 | 0.03 | 1.13 | 2.13 |
| 3.0 | 1.60 | 0.10 | 2.30 | 2.55 |
| 3.5 | 0.90 | 0.40 | 1.70 | 2.90 |
| 4.0 | 0.60 | 0.10 | 1.80 | 2.60 |
| T̄ (K) | 1.2 | 0.4 | 1.53 | 2.40 |

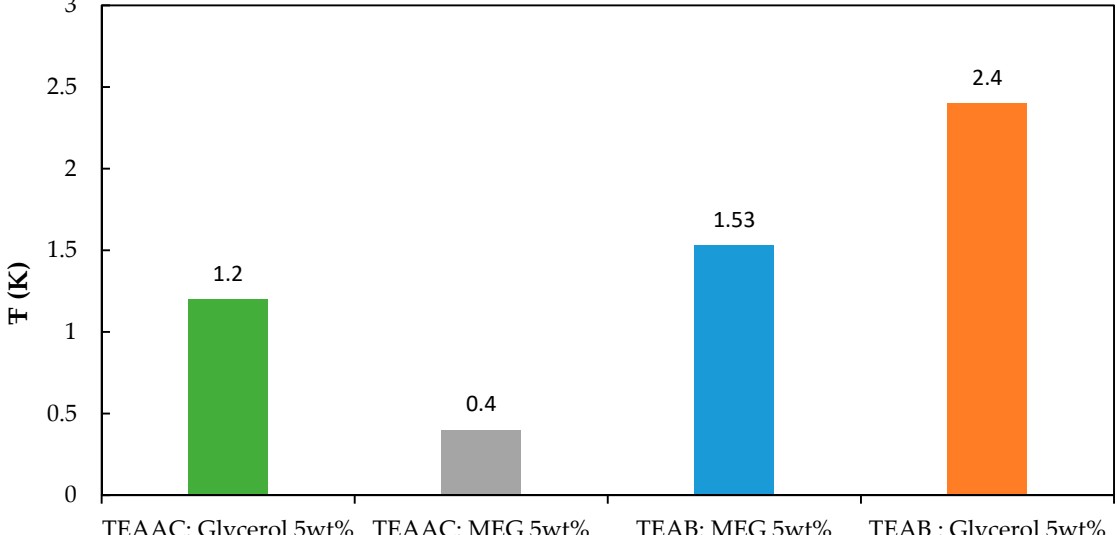

**Figure 10.** The average suppression temperature of the studied DESs.

The results proved that the TEAB:Glycerol-based DES showed thermodynamic gas hydrate inhibition property that was superior to the other conventional and ILs-based THIs selected for this study. When it involves QAS, the suppression behavior is achieved due to the hydrogen bonding of cations and anions of QASs with water molecules [14,15]. The same reasoning is given for the successful hydrate suppression shown by DES. DESs show an affinity towards water molecules to form strong hydrogen bonds with them. The naturally available high number of hydroxyl groups in DESs gives it the enormous potential to perform as THIs.

The characteristics exhibited by Choline Chloride, Propanedioic Acid, 3-Phenyl Propionic Acid, Itaconic Acid, and 3-Mercaptopropionic Acid-based DESs, as potential shale inhibitors in water-based drilling fluids, are additional evidence of the DESs to be efficient THIs [37]. In research on dehydration of natural gas by Aissaoui et al. [38], DESs based on Choline Chloride displayed significant absorption of $H_2O$ from natural gas down to the concentrations specified for pipelines to avoid hydrate formation. These works show the potential of DESs to be an ultimate alternative to the currently available conventional THIs, which have a lot of drawbacks.

### 3.3. Gas Hydrate Dissociation Enthalpies ($\Delta H_{diss}$)

The participation of inhibitors in the hydrate crystalline structure could be evaluated by measuring the gas hydrate dissociation enthalpy, $\Delta H_{diss}$ [15]. The $\Delta H_{diss}$ of the systems tested in this study is presented in Table 7.

**Table 7.** Dissociation enthalpies ($\Delta H_{diss}$) of DES systems at different pressure conditions.

| Pressure (MPa) | Pure Water | TEAAC:Glycerol 5 wt % | TEAAC:MEG 5 wt % | TEAB:MEG 5 wt % | TEAB:Glycerol 5 wt % |
|---|---|---|---|---|---|
| 2.0 | 69.60 | 66.95 | 70.02 | 77.37 | 74.47 |
| 2.5 | 66.45 | 64.42 | 67.01 | 73.71 | 70.88 |
| 3.0 | 62.96 | 62.34 | 63.64 | 69.57 | 67.04 |
| 3.5 | 59.22 | 58.68 | 59.83 | 65.44 | 62.71 |
| 4.0 | 55.58 | 56.38 | 55.86 | 60.56 | 58.18 |
| Average ($\Delta H_{diss}$) (kj/mol) | 62.76 | 61.75 | 63.27 | 69.33 | 66.66 |

Expanded uncertainties U (T) = ±0.1 K, U (P) = ±0.01 MPa, U (mass fraction) = ± 0.0001 g, U (H) = ± 1.2 kJ·mol$^{-1}$ (0.95 level of confidence).

It is well established that $CO_2$ gas hydrates form structure I hydrates [7,51–53]. The main parameters that alter the dissociation enthalpy of hydrates are the hydrogen bonding of the clathrate structure and the cage occupancy of the guest molecule [54]. The results obtained showed that the $\Delta H_{diss}$ value did not vary significantly in the presence of DES, which signifies DES molecules having no contribution in the hydrate crystallization process. As the enthalpy values were almost similar without the inhibitor, it can be deduced that THIs behavior shown by the studied DESs are mainly due to the hydrogen bonding ability.

A key advantage of DES is also the ability to solve the corrosion issues in the pipeline. It is well known the most gas hydrate inhibitors often have antagonistic behavior, which promotes corrosion while suppressing the hydrate formation [12]. The detailed corrosivity study by Ullah et al. [55] on the $CO_2$ saturated DES system and $CO_2$ saturated MEA system had a corrosion rate of 0.027 mm/year and 0.54 mm/year, respectively. The results prove that the DESs of Cholinium Chloride and Levulinic Acid are good candidates for corrosion inhibition in the fuel pipelines. There are several potentials to be exploited with the introduction of Deep Eutectic Solvents.

## 4. Conclusions

As DESs are generally formed by mixing HBD and HBA, there is a significant margin to be exploited in making the greener gas hydrate inhibitors. Many substances have been tested to be potential gas hydrate inhibitors but have shown either biodegradability or performance issues. However, mixing compounds to make DESs may be a beneficial act. In this work, the DESs were made at a 1:7 molar ratio for the combinations of TEAAC:MEG, TEAAC:Glycerol, TEAB:MEG, and TEAB:Glycerol. The DES, TEAB:Glycerol showed a significant average suppression temperature ($\bar{T}$) of 2.4. The Hydrate Liquid-Vapor Equilibrium (HLVE) data for $CO_2$ were evaluated through the T-cycle method at different temperature (273.15–283.15 K) and pressure (2–4 MPa) conditions in the presence and absence of 5 wt % aqueous DES solutions. The concentration of the best performing DES was then increased from 5 wt % to 10 wt % to observe any changes. The hydrate inhibiting behavior was also compared to other conventional and ILs-based THI, where the DES (TEAB:Glycerol) proved to be superior. It is recommended that further studies are made on DES to maximize its usage and move towards greener industry.

**Author Contributions:** Conceptualization, V.S. and B.L.; Data curation, V.S.; Formal analysis, V.S. and N.H.; Funding acquisition, Z.K.; Investigation, V.S. and N.H.; Methodology, V.S. and N.H.; Project administration, Z.K.; Resources, B.L.; Supervision, B.L. and A.S.M.; Validation, V.S.; Writing–original draft, V.S.; Writing–review and editing, V.S. and A.S.M. All authors have read and agreed to the published version of the manuscript.

**Funding:** This research received no external funding.

**Acknowledgments:** The authors would like to thank Universiti Teknologi PETRONAS, Malaysia, for the financial and technical aids provided. Special thanks to those who have contributed to this work.

**Conflicts of Interest:** The authors declare no conflict of interest.

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
