# Peer review of "Tetraethylammonium Acetate and Tetraethylammonium Bromide-Based Deep Eutectic Solvents as Thermodynamic CO2 Gas Hydrate Inhibitors"

_applsci, doi:10.3390/app10196794_

Round 1

Reviewer 1 Report

Some remarks have to be taken into account by the authors:

  • In Sec. 2.2.1 Characterization using Fourier Transform Infrared Spectroscopy (FTIR) - No description of the measurement conditions such as the resolution, the number of scans, and the temperature, type of apparatus, producer. Was KBr or ATR module used? What was the form of samples? Please add.
  • Sufficient details of methods such as type apparatus, measurement conditions, producer, should be described to allow others to replicate and build on published results.
  • Some figures have low resolution e.g. Fig. 1, etc. It provides high resolution images.
  • How many samples of one type (or the number of replicates of the experiment) were used in the swelling studies? Lack of information about the number of replicates of the experiment.
  • Error range or the determination coefficient should be included, especially Tables. Statistical analysis should be emphasized to support the results, discussion and conclusion.

Reviewer 2 Report

This manuscript by Prof. Lal et al. describes the employment of deep eutectic solvents as CO2 gas hydrate inhibitors. The author has evaluated a series of DESs composed of TEAAC/TEAB and MEG/glycerol in various ratios. The results indicate that lower temperature and higher pressure are required for the formation of CO2 hydrates when DESs are used. The author has shown that TEAB/glycerol pair is the best DES compared with other HBA/HBD pairs while the TEAB/glycerol 10 wt% is superior relative to TEAB/glycerol 5 wt%. Given the results, this reviewer recommends the author to add the data acquired with TEAB/glycerol 10 wt% to the manuscript in addition to 5 wt%. All the thermodynamic results shall be included in the manuscript to complete it.
